# A Preliminary Study on Cranio-Facial Characteristics Associated with Minor Neurological Dysfunctions (MNDs) in Children with Autism Spectrum Disorders (ASD)

**DOI:** 10.3390/brainsci10080566

**Published:** 2020-08-18

**Authors:** Laura Maniscalco, Bonnet-Brilhault Frédérique, Michele Roccella, Domenica Matranga, Gabriele Tripi

**Affiliations:** 1Department of Biomedicine, Neuroscience and Advanced Diagnostics-BIND-University of Palermo, 90127 Palermo, Italy; laura.maniscalco04@unipa.it; 2UMR 1253, iBrain, Université de Tours, Inserm, 37000 Tours, France; frederique.brilhault@univ-tours.fr; 3Department of Psychological Sciences, Pedagogical and Education, University of Palermo, 90128 Palermo, Italy; michele.roccella@unipa.it; 4Department Promozione della Salute, Materno-Infantile, di Medicina Interna e Specialistica di Eccellenza, “G. D’Alessandro”-PROMISE-University of Palermo, 90127 Palermo, Italy; domenica.matranga@unipa.it; 5Department of Childhood Psychiatry for Neurodevelopmentals Disorders, Centre Hospitalier du Chinonais, 37500 Saint-Benoît-la-Forêt, France

**Keywords:** autism spectrum disorders, morphology, minor neurological dysfunctions, neurodevelopment

## Abstract

**Background.** Craniofacial anomalies and minor neurological dysfunction (MNDs) have been identified, in literature, as risk factors for neurodevelopmental disorders. They represent physical indicators of embryonic development suggesting a possible contributory role of complications during early, even pre-conceptional, phases of ontogeny in autism spectrum disorders (ASD). Limited research has been conducted about the co-occurrence of the two biomarkers in children with ASD. This study investigates the associative patterns of cranio-facial anomalies and MNDs in ASD children, and whether these neurodevelopmental markers correlate with intensity of ASD symptoms and overall functioning. **Methods.** Caucasian children with ASD (n = 33) were examined. Measures were based on five anthropometric cranio-facial indexes and a standardized and detailed neurological examination according to Touwen. Relationships between anthropometric z-scores, MNDs and participant characteristics (i.e., age, cognitive abilities, severity of autistic symptoms measured using the Childhood Autism Rating Scale (CARS) checklist) were assessed. **Results.** With respect to specific MNDs, significant positive correlations were found between Cephalic Index and Sensory deficits (*p*-value < 0.001), which did not correlate with CARS score. Importantly, CARS score was positively linked with Intercanthal Index (*p*-value < 0.001), and negatively associated with posture and muscle tone (*p*-value = 0.027) and Facial Index (*p*-value = 0.004). **Conclusion.** Our data show a link between a specific facial phenotype and anomalies in motor responses, suggesting early brain dysmaturation involving subcortical structures in cerebro-craniofacial development of autistic children. This research supports the concept of a “social brain functional morphology” in autism spectrum disorders.

## 1. Introduction

Over the past few years, there has been mounting evidence to support the neurodevelopmental model of autism spectrum disorders (ASD), which postulates that the etiological origins of the disease can be traced to events in the prenatal period [1].

Several studies have investigated brain structure and brain function in subjects with neurodevelopmental impairments including; autism spectrum disorders (ASD), ADHD and schizophrenia. These studies identified abnormal brain structure and a higher incidence of developmental brain anomalies, resulting in clinical, cognitive, electrophysiological and neuroanatomical markers. [2,3]. Among these so-called “neurodevelopmental” markers, minor physical abnormalities (MPAs) and minor neurological dysfunction (MNDs) have been suggested as risk factors for ASD and, correspondingly, as sensitive physical indicators of embryonic development [4,5,6].

MPAs are neurodevelopmental markers which manifest as unusual morphological features of the face or physique. MPAs can manifest as early as the first or early second trimester, and have consistently been linked to a biological vulnerability, which continues into adolescence and young adulthood [5,6,7].

MPAs associated with ASD, have been observed across multiple areas of the body. In particular, re-occurrence of anomalies in the craniofacial region, suggests specific facial characteristics related to subpopulations of ASD. There has been consistent evidence that the development of the face is intimately entwined with development of the brain; their tissues share common origins, and develop in close coordination, as a result of physical adjacency and reciprocal molecular signaling [8].

Previous studies [9,10,11] have shown that facial morphology differed significantly in individuals with ASD diagnoses; identifying subsets of ASD boys, with distinctive craniofacial morphology, associated with greater severity of ASD symptoms. This could suggest an “ASD facial phenotype”, representing a new physical biomarker which could be used to improve ASD diagnosis.

The concept of minor neurological dysfunction (MND) can be useful when describing neurodevelopmental impairments in ASD children. In recent years, attention has focused upon sensory-motor abnormalities observed in subjects with ASD, categorizing these as “associated symptoms”. Infants and children with ASD have been found to display altered motor development, clumsiness, retention of primitive reflexes, deficits in gross and fine motor movement, impaired postural control and abnormal gait sequencing [12,13,14].

These dysfunctions have neurodevelopmental relevance, indicating specific alterations in the connecting fiber systems of the central nervous system, which result in impaired sensory-motor performance. An adverse event at any stage of neurological development may result in various clinical manifestations of MNDs, at different life stages [14]. Two etiologically and clinically different forms of MNDs can be distinguished: simple and complex MNDs. Simple MND may be regarded as an expression of typical, but non-optimal brain wiring. In contrast, complex MND denotes clinically relevant brain dysfunction and is considered to be a borderline form of cerebral palsy [14]. In the literature, children with MNDs show higher rates of MPAs than children without MND, suggesting a possible contributory role of complications during early, even pre-conceptional, phases of ontogeny [14,15,16,17].

Despite evidence to suggest a link between minor neurological dysfunctions and minor anomalies of the face and limbs, in neurodevelopmental conditions such as schizophrenia [18] or developmental coordination disorders [14], comparatively little is known about the co-occurrence of the two biomarkers in children with ASD.

The aim of the present study was to investigate the associative patterns of cranio-facial anomalies and MNDs in ASD children, and to determine whether these neurodevelopmental markers correlate with intensity of ASD symptoms and overall functioning. More generally, assessment of MPAs and MNDs may assist in the diagnostic protocol for neurodevelopmental disorders.

## 2. Methods

### 2.1. Participants

A total sample of 33 children over a total population of 65 individuals with ASD diagnosis attending at Tours University Hospital Center (Tours, France) was selected. The subjects’ selection is showed in Figure 1. The sample consisted of Caucasian children of European origin, as normative physical data have been well-defined in the Caucasian population [9,10]. The exclusion criteria regard non-Caucasian ASD children, ASD children with clinical diagnosis of a co-morbid psychiatric or medical condition such as epileptic seizures or frank neurological pathology (e.g., cerebral-palsy, muscular dystrophy or evidence of other) and ASD children in pubertal age. The latter was an exclusion criterion since puberty is associated with a substantial decline in the number of dysfunctional clusters of MND. All subjects met the current ICD-10 criteria for clinical diagnosis of “pervasive developmental disorders (PDD)” according to the World Health Organization’s (WHO) International Classification of Diseases, 10th edition (ICD-10). ASD was diagnosed using standardized diagnostic tools (mean age at assessment: 7.6 years; standard deviation SD: 2.2). The sample of ASD children included those who met diagnostic criteria for; childhood autism (F84.0, n = 21), atypical autism; (F84.1, n = 9), Asperger’s syndrome (F84.5, n = 2) and other PDD (F84.9, n = 1). ADI-R [19] complemented the clinical diagnosis of ASD children. One child classified as having PDD not otherwise specified on the ADI-R (below threshold on 1 domain by 1 point) was also included.

The study was conducted in accordance with the basic principles of the Declaration of Helsinki. Clinical data, including behavioral and cognitive evaluation, were collected from medical records, stored in a bio-clinical database. This was approved by CNIL, the French data protection authority.

An extensive, multidisciplinary child neuro-psychiatric assessment was carried out for each subject. Assessment consisted of a preliminary developmental history, medical examination and neuropsychological assessment.

Behavioral assessment was carried out using the Childhood Autism Rating Scale (CARS) [20]. The CARS evaluates the severity of autistic behaviors in 14 functional areas by assigning a score from 1 to 4. An overall score is calculated by adding scores from all functional areas. It enables the assessment of severity of autistic symptoms, and the stratification of patients into three levels: “severely autistic” (score between 37 and 60), “mildly to moderately autistic” (score between 30 and 36.5), and “absence of ASD” (score less than 30). The questionnaire can be carried out in around 20–30 min. Clinical psychologists were trained on these scales and were experienced with children with ASD. Intellectual functioning was evaluated using the Weschler Abbreviated Intelligence Scale-III, (WAIS III) [21], allowing a global Intellectual Quotient (IQ), a non-Verbal Quotient (nVQ) and a Verbal Quotient (VQ) to be obtained for all ASD children.

All ASD children received a comprehensive physical examination, to exclude the presence of a clinically detectible genetic syndrome associated with gross dysmorphic features and autistic-like symptoms. The examination was conducted by a pediatric team, in the same hospital site, examiners were blind to group status. Written informed consent was obtained from the parents or guardians of each participant.

### 2.2. Morphological Assessment

For a detailed description of the morphological assessment protocol see Tripi et al. [11] Cranio-facial examination of the ASD group employed a mixed approach of computerized photogrammetry and classic anthroposcopy. The examination was conducted by the same author (GT) in the same testing rooms. Photographs were obtained using a high-resolution digital camera. The Euclidean measurements, based on anthropometric landmarks and 2D photogrammetric examinations, were obtained using image analysis software (FlashCAD^®^, Digitarch srl, Rome, Italy) [6]. Standardized cranio-facial photographs were taken with the camera lens aligned with the subject’s Frankfort horizontal plane [22]. An internal measure of scale (adhesive paper sticker) was placed on the glabella landmark for the frontal view, and on the condylion of the mandible for the profile view.

Anthropometric linear measurements and landmarks on the soft tissue of the face and head, were based on five cranio-facial indexes, borrowed from the works of Farkas et al. [23]. Each skull-facial index was compared with the tables of anthropometric norms for Caucasian subjects.

### 2.3. Neurological Assessment

All ASD children underwent a standardized and detailed neurological examination according to Touwen [13]. This is a standardized and age-specific examination, which focuses on minor neurological dysfunctions. The following clusters were evaluated: posture and muscle tonus; reflexes; involuntary movements; coordination and balance; fine motor movements; associated movements; sensory functions and cranial nerve function. The assessment was conducted by the same author (GT), in the same testing rooms, over a duration of one hour. The assessment criteria used to identify MNDs was age specific; as developmental changes in the nervous system are known to induce changes in the expression and prevalence of MNDs. Literature data [14] indicate that the rate of MNDs at preschool age in the general population is relatively low, reaching its peak before the emergence of puberty. Signs of dysfunction are taken into account only if they occur in cluster. The presence of a single sign of dysfunction, such as isolated positive Babinski signs, does not lead to a diagnosis of MND. Clusters, as defined in clinical practice, are organized according to the functional neuro-behavioral subsystems of the nervous system [24].

The original protocol suggested by Touwen [13] was adapted slightly. Adaptation consisted of removal of: following an object with rotation of the trunk whilst sitting, palmo-mental reflex, Mayer and Leri reflexes, cremasteric reflex, Galant response, examination of the spine whilst the child is lying down, examination of the hip joints, sitting up without the help of hands, fundoscopy and localization of sound. These items were omitted as some authors [25] suggest they have little value in terms of determining the presence or absence of clusters of MNDs in the general population.

According to the classification of Hadders-Algra [14,16,17], ASD children were classified as neurologically normal if having no abnormal domains. If one or two abnormal domains were classified as s-MND and three or more abnormal domains as c-MND.

### 2.4. Statistical Methods

Statistical analyses were carried out using R software (version 3.6.3) available at https://cran.r-project.org/. Categorical variables were summarized as counts and percentages, discrete and continuous variables as mean, standard deviation (SD) and range. Z-scores of Cranio-facial Indexes were calculated using tables of anthropometric norms for Caucasian subjects, developed by Farkas et al. [23]. Relationships between anthropometric index (cephalic, facial, intercanthal, nasal and mouth-face) z-scores and participant characteristics (age, sex, global QI, VQ-nVQ, severity of autistic symptoms measured using the CARS score) and MNDs (posture and muscle tone, reflex abnormalities, involuntary movements, coordination and balance, fine motor dysfunction, associated movements, sensory deficits, cranial nerve dysfunction) were assessed using Pearson’s correlations. A standard multiple regression model was applied to assess craniofacial Index z-scores (as response variables) in relation to participant characteristics and MNDs (as independent variables), adjusted for participant characteristics. A linear regression was applied to explain CARS score in relation to demographic variables, craniofacial Indexes, MNDs, net of global QI. Results reporting *p*-value < 5% were considered statistically significant.

## 3. Results

The study sample included 33 children, (28 of them are males), aged 91.5 (26.7) months with a range of 4 and 12 years of age. Demographic, clinical and anthropometric characteristics of the sample are shown in Table 1. The prevalence of MNDs in our sample was 91%, of which 17 children (57%) were affected by simple MND and 13 children (43%) had complex MNDs. The most frequent minor neurological dysfunction was associated movements (58%), and more than half of the sample suffered from sensory deficits (52%). Fine-motor dysfunction was present in 33% of the sample whilst posture and muscle tone and coordination and balance were affected in 30% of children (Table 2).

The correlations between anthropometric z-scores, demographic characteristics, global QI, CARS and MND are shown in Table 3. Most of these correlations were previously described in our study [11]. Moreover, significant, large positive correlations were found between cephalic index and sensory deficits (r = 0.547, *p*-value < 0.001) and between cephalic index and c-MND (r = 0.469, *p*-value = 0.005). Conversely, a negative correlation was discovered between Cephalic Index and s-MND (r = −0.448, *p*-value = 0.008). Facial index is significantly, negatively correlated with CARS score (r = −0.505, *p*-value = 0.002) and Sensory deficits (r = −0.532, *p*-value = 0.001), and positively correlated with complex MND (r = 0.468, *p*-value = 0.005). While a moderate correlation was observed between facial index and global QI (r = 0.348, *p*-value = 0.046). Furthermore, moderate correlations were discovered between intercanthal index and CARS score (r = 0.384, *p*-value = 0.027) and reflex abnormalities (r = 0.381, *p*-value = 0.028) (Table 3).

Since CARS score has previously been linked to intellectual functioning (Militerni 2002 et al. [26] in our study, r = 0.683, *p* < 0.001), standard multiple regression analyses were performed to clarify the relationships between anthropometric z-scores, as dependent variables, and demographic characteristics; global QI, CARS and MNDs (Table 4). For the sake of simplicity, only coefficients with significant or borderline *p*-values were reported. The regressions’ results confirm that the cephalic index was positively correlated with sensory deficits (coeff = 1.10, *p*-value = 0.013) (Table 4). Moreover, facial index is negatively correlated with CARS score (*p*-value = 0.035), for every one-unit increase in CARS score, the facial index z-score decreased by 0.118. The presence of sensory deficits is associated with a decrease of 1.332 in the z-score of facial index (*p*-value = 0.005) (Table 4).

The standard linear regression with CARS as dependent variable, showed that, an increase in the intercanthal index was linked to an increase in CARS score (*p*-value < 0.001); for every one-unit increase in the intercanthal index, the CARS score increased by 1.877. CARS score was negatively associated with posture and muscle tone (coeff = −3.155, *p*-value = 0.027) and acial Index (*p*-value = 0.004), for every one-unit increase in the facial index, the CARS score decreased by 1.410 (Table 5).

Figure 2 represents the results of the linear regression with CARS score as dependent variable. In the 3D plot, the red points represent children with dysfunction in posture and muscle tone; black points represent children without dysfunction in posture and muscle tone. The corresponding linear planes were created with the linear regression coefficients computed fixing the level of global QI at 60—the mean value in our sample.

The results of the linear regression with global QI as response variable, revealed that it was negatively correlated with MND (coeff = −11.646, *p*-value < 0.001) and positively correlated with the intercanthal index (coeff = 8.805, *p*-value = 0.002), net of the CARS score. However, an increase in the number of minor neurological dysfunction is associated with a lower global QI, while an increase in the intercanthal index is associated with an increase in the global QI (Table 5).

## 4. Discussion

Children with ASD diagnoses comprise a heterogeneous population with significant variation in; type, number and severity of social deficits, behavior, communication, and cognitive difficulties; undoubtedly reflecting multiple etiologic origins. An initial step towards identifying etiologically discrete autism subgroups, is the discovery of discrete, quantifiable and patho-physiologically relevant phenotypic features present in some, but not all, ASD subjects [8].

In the literature, MPAs and MNDs have been identified as putative biomarkers in providing insight into early neurodevelopment [5,6,14,15]. The causes of autism may be (epi) genetically determined, mildly associated with gene activity in early development, or result from prenatal environmental risk factors [27]. This suggests that a putative biomarker may well have neurodevelopmental origins. Furthermore, craniofacial anomalies and sensorimotor impairments are strongly associated with structural abnormalities in the brain, which are correlated with clinical severity in ASD [14,15,28].

Recent studies [9,10,11] showed that children diagnosed with ASD have a distinctive facial phenotype, which correlates with symptom severity, suggesting a biological and anatomical subset of ASD.

Similarly, MNDs may indicate, disruption to fronto-striatal pathways and basal ganglia, as well as alterations in the cerebellar region and brain stem. However, no particular sensorimotor symptomatology is uniquely identified as ASD [29,30].

To our knowledge no previous study has examined the relationships between ASD facial phenotype and MNDs.

Our recent research [11] identified a dolichocephalic head shape in ASD children, which did not correlate with autism severity. This finding was compatible with previous research, showing that brain and head size is a relatively non specific finding in autism [27] and suggesting an abnormal brain growth trajectory during the in-utero period, after the 22nd week of amenorrhea, in children with ASD who showed postnatal head overgrowth [31].

The current study found a direct correlation between dolichocephalic head shape and c-MND and an inverse correlation with s-MND. Complex MND can be considered as a distinct form of perinatally acquired brain dysfunction, which is likely to be associated with a structural deficit in the brain [14]. Moreover, with respect to specific MNDs, dolichocephaly head shape is associated with sensory anomalies, but does not correlate with autism severity.

Frequently occurring types of neurological dysfunction, can been linked to specific parts of the brain. They suggest dysfunction in cortical structures (dysfunction in fine manipulative abilities), in cerebellar–brainstem circuitries (sensory deficits and dysfunctional regulation of posture and muscle tone and coordination), and more global dysfunction (excess of associated movements, which may point to immature or inadequate pathways and/or requiring more effort to perform movements) [32]. The link between dolichocephalic head shape and sensory deficits in our ASD group, could suggest a strong contribution of dysfunction of subcortical structures in brain growth trajectory. However, the finding that an increase in the number of MNDs is associated with lower global QI, represents a limitation in the generalizability of our result, and adds to the evidence [33] for an independent association between minor neurological impairment and cognitive performance.

On the other hand, our previous research [11] identified two craniofacial characteristics directly associated with autism severity; increased inter-orbital distance and reduced height of the facial midline. This ASD facial phenotype, supports the concept of a tight orchestration between atypical processes governing the development of the “social brain” subcortical structures (such as the amygdala) during gestation, and perturbations in the midline structures of the human embryonic face [7,9].

Our new data show a link between this ASD facial phenotype and anomalies in posture and muscle tone, directly associated with autism severity.

In the literature the limbic structures represent key neural candidates, to orchestrate the rapid integration of emotional inputs and motor output processes [34]. Recent findings suggest that, in humans, the amygdala promotes protective motor reactions in emotionally-significant contexts, and influences the execution of ongoing actions, by modulating functional interactions with supplementary motor areas and interconnected motor pathways [35].

A recent study [36], has shown that the amygdala is altered in ASD, not only in terms of its volume, but also in the microstructural properties of its connections to the cortex. These volumetric and structural anomalies are associated with increased severity of the autistic phenotype, as measured by impaired emotion recognition. Moreover, amygdala sub-regions may have specific associations with ASD traits.

So, the link between this specific facial phenotype and anomalies in motor responses, could be suggestive of early brain dysmaturation involving the amygdala, in cerebro-craniofacial development, in ASD. This suggests a link to “social brain” functional (dys)morphology in ASD.

Intriguingly, our analysis also shows that the presences of sensory deficits, which are strongly associated with ASD clinical criteria, are linked to a decrease in the height of the facial midline. This finding adds support to the pivotal role of subcortical structures in cerebro-craniofacial development of autistic children.

Several limitations can be identified in the current study. The primary limitation was a lack of genetic data. ASD is a heterogeneous disorder; it does not have a single physical phenotype which would be suggestive of an underlying genetic syndrome. We excluded ASD children with comorbid gross dysmorphic features, due to the direct association in literature [37] between the severity of physical anomalies and increased likelihood of finding cytogenetic alterations.

Indeed, genetic testing methods, such as high-resolution karyotype and chromosomal microarray, are now considered first-tier testing for individuals with ASD, developmental delay, and multiple congenital anomalies. However, these are not universally obtained by medical providers in clinical practice [37].

Similarly, we did not obtain neuroimaging data for our ASD subjects, but the results of our study add to findings from existing neuroimaging data identifying abnormalities in; basal ganglia, thalamus, supplementary motor areas and cerebellum; as a likely source of sensory-motor impairment in autism spectrum disorder [24,37].

Our results could have clinical relevance for evaluation and treatment of children with ASD and other Neurodevelopmental disorders. Children with a distinctive craniofacial phenotype (e.g., fetal alcohol spectrum disorders), who lack the classic facial phenotype on external exam, but who have significant differences in shapes and volumes, may be considered candidates for further neuroimaging investigations of brain abnormalities and underlying genetic disorders [38].

The second limitation of this study was its relatively small sample size. Moreover, the study population was limited to Caucasian patients, as ethnicity can influence the prevalence of morphological abnormalities. Future studies are required to establish similar norms for other ethnic groups.

Thirdly, our methodology of 2D photogrammetry is limited to Euclidean measurement-although it has the advantage of being a significantly less expensive approach than 3D measurement techniques.

In conclusion our work suggests a direct association between craniofacial development and specific neurological anomalies in ASD; and supports preliminary the concept of a “social brain” functional (dys)morphology in autism spectrum disorders.

## Figures and Tables

**Figure 1 brainsci-10-00566-f001:**
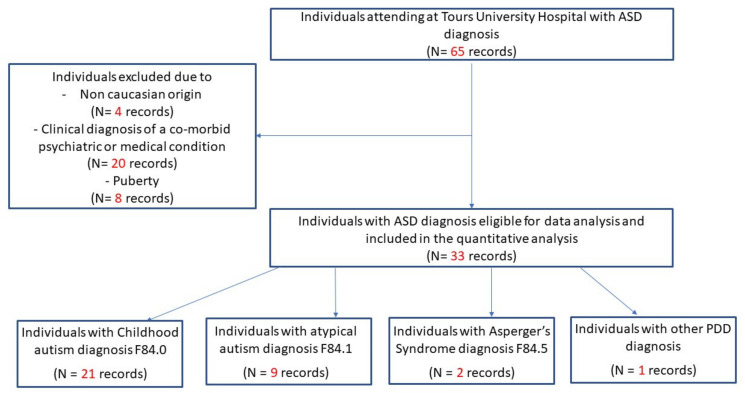
Flow-chart diagram of subjects’ selection.

**Figure 2 brainsci-10-00566-f002:**
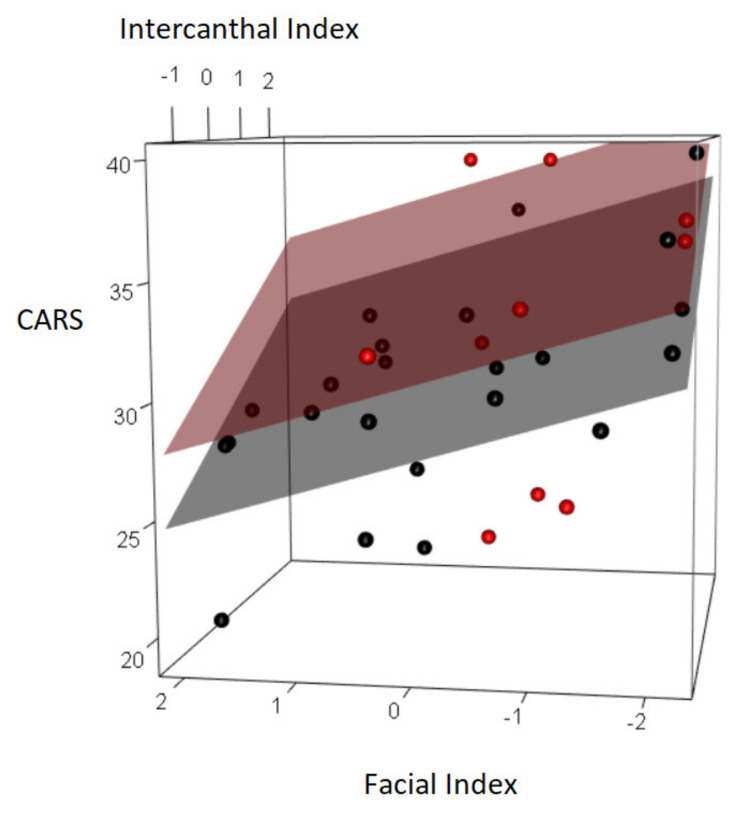
3D plot representing the coefficients of linear regression analysis on CARS as dependent variable.

**Table 1 brainsci-10-00566-t001:** Demographic, clinical and anthropometric characteristics of the 33 children with ASD.

Demographic and Clinical Variables		Anthropometric Variables		
	Mean (SD)		Mean (SD)	Z-Score
Age in months	91.5 (26.7)	Cephalic Index	74.2 (5.1)	−0.51 (1.2)
CARS	30.4 (5.3)	Facial Index	83.9 (6.6)	−0.17 (1.3)
Global QI	59.9 (24.2)	Intercanthal Index	38.7 (2.2)	0.20 (1.0)
VQ	50 (24.4)	Nasal Index	72.5 (5.9)	0.27 (0.9)
nVQ	70 (26.8)	Mouth-Face Index	36.7 (2.1)	−0.09 (0.9)
MND Total	2.2 (1.5)			

**Table 2 brainsci-10-00566-t002:** Prevalence of the specific types of MND in ASD children.

Type of Mild Neurological Dysfunction	
	n (%)
Posture and muscle tone	10 (30%)
Reflex abnormalities	1 (3%)
Involuntary movements	3 (9%)
Coordination and balance	10 (30%)
Fine motor dysfunction	11 (33%)
Associated Movements	19 (58%)
Sensory deficits	17 (52%)
Cranial nerve dysfunction	3 (9%)
s-MND	15 (45%)
c-MND	14 (42%)

**Table 3 brainsci-10-00566-t003:** Correlations between anthropometric z-scores and age at assessment, CARS, and IQ.

	Cephalic Index Z-Score		Facial Index Z-Score		Intercanthal Index Z-Score		Nasal Index Z-Score		Mouth-Face Index Z-Score	
	r	*p*-Value	r	*p*-Value	r	*p*-Value	r	*p*-Value	r	*p*-Value
Age	0.330	0.060	−0.061	0.732	0.143	0.426	−0.044	0.807	0.001	0.999
CARS	0.315	0.073	−0.505	0.002	0.384	0.027	−0.274	0.122	−0.029	0.870
Global IQ	−0.231	0.194	0.348	0.046	−0.121	0.499	0.137	0.444	0.074	0.678
VQ	−0.290	0.101	0.316	0.072	−0.253	0.154	0.119	0.506	0.058	0.747
nVQ	−0.182	0.308	0.328	0.061	0.012	0.944	0.208	0.244	0.036	0.838
Posture and muscle tone	0.432	0.012	−0.293	0.097	0.278	0.116	−0.196	0.272	0.192	0.283
Reflex abnormalities	0.186	0.299	0.049	0.785	0.381	0.028	−0.259	0.145	0.174	0.332
Involuntary movements	0.042	0.812	−0.145	0.419	0.276	0.118	0.184	0.304	−0.205	0.252
Coordination and balance	0.203	0.255	−0.081	0.652	0.132	0.461	−0.233	0.191	−0.015	0.933
Fine motor dysfunction	−0.094	0.609	0.011	0.948	0.296	0.096	−0.093	0.604	−0.322	0.067
Associated Movements	0.158	0.378	−0.298	0.091	0.154	0.391	−0.021	0.906	−0.048	0.788
Sensory deficits	0.547	<0.001	−0.532	0.001	0.064	0.720	−0.088	0.626	0.206	0.248
Cranial nerve dysfunction	0.034	0.846	−0.015	0.931	0.175	0.329	0.278	0.116	−0.099	0.581
s-MND	−0.448	0.008	0.271	0.126	−0.244	0.170	0.118	0.511	−0.083	0.644
c-MND	0.469	0.005	−0.468	0.005	0.245	0.168	−0.150	0.403	0.013	0.941

**Table 4 brainsci-10-00566-t004:** Standard multiple regression analyses on anthropometric z-scores as dependent variables and demographic characteristics, global QI, CARS and MND.

	Coef	95%CI		*p*-Value
**Cephalic Index**				
Intercept	−5.455	−10.786	−0.125	0.045
Fine motor dysfunction	−0.899	−1.827	0.028	0.056
Sensory deficits	1.100	0.253	1.948	0.013
**Facial Index**				
Intercept	4.017	−1.591	9.626	0.150
CARS	−0.118	−0.227	−0.009	0.035
Reflex abnormalities	2.070	−0.355	4.497	0.090
Sensory deficits	−1.332	−2.225	−0.441	0.005
**Intercanthal Index**				
Intercept	−4.229	−9.392	0.933	0.102
Global IQ	0.025	−0.001	0.051	0.063
**Mouth-Face Index**				
Intercept	0.016	−5.156	5.188	0.995
Fine motor dysfunction	−0.842	−1.742	0.059	0.065

Coef = Regression coefficients; 95%CI = 95% Confidence interval; The bold is for a better understanding of anthopometrical variables.

**Table 5 brainsci-10-00566-t005:** Standard linear regression analyses on CARS and Global QI as dependent variables.

	Coef	95%CI		*p*-Value
**CARS**				
Intercept	39.035	35.286	42.784	<0.001
Global QI	−0.138	−0.191	−0.085	<0.001
Posture and muscle tone	−3.155	−5.944	−0.367	0.027
Facial Index	−1.410	−2.338	−0.482	0.004
Intercanthal Index	1.877	0.751	3.004	0.001
**Global QI**				
Intercept	162.072	126.819	197.326	<0.001
CARS	−2.674	−3.894	−1.455	<0.001
Total MND	−11.646	−16.631	−6.663	<0.001
Coordination and balance	9.262	−3.752	22.278	0.155
Facial Index	−4.016	−8.620	0.587	0.084
Intercanthal Index	8.805	3.281	14.331	0.002

Coef = Regression coefficients; 95%CI = 95% Confidence interval.

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
