# Peer review of "A Preliminary Study on Cranio-Facial Characteristics Associated with Minor Neurological Dysfunctions (MNDs) in Children with Autism Spectrum Disorders (ASD)"

_brainsci, 2020, doi:10.3390/brainsci10080566_

Round 1

Reviewer 1 Report

This is a thoughtfully written study that investigates the association between cranio-facial characteristics and Minor Neurological Dysfunction in Children with Autism SPectrum Disorders.  The authors present a well developed literature review on the background of facial dysmorphology and neurodevelopmental markers.  The authors discuss previous research in facial dysmorphology and ASD.  There is a significant amount of literature in other neurodevelopmental disorders that are associated with facial dysmorphology, specifically Fetal Alcohol Spectrum Disorders, that could be referenced and serve as a comparison group for this research direction. 

Despite the small sample size, which is appropriately addressed as a limitation of the study, the study is well designed and encompasses an adequate assessment of a range of neurodevelopmental characteristics,  facial measurements, and severity of ASD symptomatology.

The discussion section identifies that an initial step towards identifying etiologically discrete autism subgroups is the discovery of discrete, quantifiable and patho-physiologically relevant phenotype features present in some, but not all ASD subjects.  The current study contributes to this literature by demonstrating a potential link between ASD facial phenotype and anomalies in posture and muscle tone that are directly associated with autism severity.  The authors make a pretty large jump tying in their findings of specific facial phenotype and anomalies in motor responses to early brain dysmaturation involving the amygdala.  They do acknowledge that an increase in MNDs may be associated with lower global IQ and may provide a limitation to generalizability in the results. 

Overall, this is an interesting, well written paper that presents the foundation for a number of directions of future research and continuing to investigate facial dysmorphology as a potential marker of later neurodevelopmental features associated with ASD.  

Author Response

We thank the Reviewer for appreciating the paper that in according with his judgment is “an interesting, well written paper that presents the foundation for a number of directions of future research”. Regarding his comment about other neurodevelopmental disorders  associated with facial dysmorphology, (specifically Fetal Alcohol Spectrum Disorders), the recent article of Del Campo M & Jones KL 2017  was very useful for the global vision of our manuscript and was cited in the text : Discussion- Line 342-346 and References line 455-456

* English language and style of manuscript was revised by Mrs Suzanne Heron-  Proof Reading

Reviewer 2 Report

I welcome this paper as it tackles an issue that has received little attention in the field of Autism Spectrum Disorders, the namely specific facial phenotype of ASD in the context of Caucasian ethnical morphology.   There are some clear messages to be drawn from the authors' experiences, but I felt that the paper should identify some points more clearly.  Here are some examples of what I mean by clarification.

Foremost I would question the applicability of ICD-10 in the classification of ASD while ADI-R applied for the diagnosis which is based on DSM classification. There are diagnostic classifications that are not applicable and updated terms are substituted among experts in this field.

I also question the applicability of CARS for determining the level of severity while as a certified ADI-R administrator I recall that the scale is capable of determining the different types of ASD which are generally based on the level of severity of the symptoms? This is logical for an individual who classified as having Asperger's syndrome to be determined as having a less severe form of ASD.

This will be very unlikely for an individual to be diagnosed as Asperger's and classified in "severely autistic".

No data regarding individuals' language ability level reported. Although on lines 114 and 115 it is cited that "global Intellectual Quotient (IQ), a non-verbal IQ (performance IQ) and a verbal IQ to be obtained for all ASD children". This might be an indicator that the entire recruited sample was verbal, hence this is wreath to be mentioned clearly that the sample was all verbal.

If the entire recruited sample were verbal, there will be a very big question will arise regarding the level 2 and 3 of individuals with ASD that are generally nonverbal or at least reluctant to talk.

I was also very curious to know if any individual excluded because of the exclusion criteria form the potential population.  Because on line 148, it is also cited that some individuals were excluded (nor mentioned how many) because they entered into the puberty period which indicates that the sample was bigger than what is reported. Also, online number 160, it is cited that Neurological abnormality was also considered as the exclusion criteria, and based on the Touwen examination some individuals classified as neurologically abnormal but no numbers were presented. It also indicated that the originally recruited sample was much bigger than the final 33. The dropout rate also will be very useful to report.

I could not figure out the sample recruitment approach in the methods section, I speculate a convenient sampling approach considered. It needed to be mentioned although if any other approach for sample recruitment is followed.

ADI-R is a gold standard diagnostic scale with very valuable data regarding the early developmental trajectory information motor milestones and level of language abilities and loss or acquisition of different skills, why it is completely ignored in the analysis while it was available for you and you already used this scale for the diagnosis and the data was collected?

I also found that individuals with ASD have been addressed as "patient(s)" in this paper several times; this is not in line with available literature in this field. Except for line 89 in which I recommend Sample to be used as a substitution, I suggest substituting individuals /people with ASD instead of ASD patients (line 89, 110, 315, and 132).

I also suggest the “social brain functional morphology” which is mentioned in the abstract and the final line of the discussion part to be explained in paragraph at the Methods section.

I was also curious to know if the anthropometric norms for different ethnicity might differ from what is reported to be as a norm for the Caucasian or at least some explanation about any other available norms for the other ethnical group.

In sum, I understand the challenges of doing a multidisciplinary study such as what is reported here and I know how much work goes into it. Meanwhile, I also worry about the possible misinterpretations of findings of such small scale studies on a convenient sample and a special ethnical group of children with a condition in which very little is known about its causes and etiology.  Therefore, as a safeguard, I do recommend the study to be classified as a preliminary one in title to provide a base for more robust studies in terms of sample size and coverage.

A preliminary study on Cranio-Facial Characteristics associated with Minor Neurological Dysfunctions (MNDs) in Children with Autism Spectrum Disorders (ASD)

Author Response

I welcome this paper as it tackles an issue that has received little attention in the field of Autism Spectrum Disorders, the namely specific facial phenotype of ASD in the context of Caucasian ethnical morphology.   There are some clear messages to be drawn from the authors' experiences, but I felt that the paper should identify some points more clearly.  Here are some examples of what I mean by clarification.

We thank the Reviewer for his comment highlighting this important but probably underestimated topic in the research field of ASD. 

Foremost I would question the applicability of ICD-10 in the classification of ASD while ADI-R applied for the diagnosis which is based on DSM classification. There are diagnostic classifications that are not applicable and updated terms are substituted among experts in this field.

We thank the Reviewer for this observation. Of course diagnosis for autistic spectrum disorders is a medical diagnosis.  Even if ICD 11 has been recently published, nosography and diagnosis criteria are still based, in European hospitals, on the current ICD 10 classification (Line 102) . This obviously means “Pervasive Developmental Disorders”.  ASD was diagnosed using standardized instruments in all subjects. They met the current ICD-10 (WHO 1993) criteria for “pervasive developmental disorders”. Clinical diagnosis of ASD was complemented by using ADI-R (P. Partecipants - Line 108). It was used to distinguish autistic disorder from non-autistic pervasive developmental disorder.

ADI-R is a gold standard diagnostic scale with very valuable data regarding the early developmental trajectory information motor milestones and level of language abilities and loss or acquisition of different skills, why it is completely ignored in the analysis while it was available for you and you already used this scale for the diagnosis and the data was collected?

We thank the Reviewer for this observation.  The ADI-R is a diagnostic assessment tool based on the description and history of the ASD children that analyses his/her development in three areas: quality of social interactions, communication and language, and restricted interests and stereotyped behaviours . The ADI-R has good reliability and validity and is currently considered the gold standard for diagnosing children with autism for research purposes. Parents are often very informative because they are close observers of their child’s behaviours. Previous studies (Pilowsky et al. 1998) have evaluated a good agreement between the CARS and ADI-R. On the other hand the study by Ventola et al. (2006) indicated that there was not a significant agreement between ADI-R and CARS classifications (κ = 0.095, p = 0.486).  Both the CARS cutoff of 30 and stringent ADI-R criteria (meeting threshold in all three behavior domains) allow only a diagnosis of autism, and exclude some children who would be classified as the milder autism spectrum or the previously used classification of PDD-NOS (de Bildt et al. 2004). The difference in agreement may also be explained by the ADI-R relying on parents’ recollection of behaviors of children, while the CARS integrate history and direct observation. Our aim was to evaluate the associative pattern of  cranio-facial anomalies from mild to severe ASD behavio , then  CARS score was our choice for behavioral evaluation. 

I also question the applicability of CARS for determining the level of severity while as a certified ADI-R administrator I recall that the scale is capable of determining the different types of ASD which are generally based on the level of severity of the symptoms? This is logical for an individual who classified as having Asperger's syndrome to be determined as having a less severe form of ASD. This will be very unlikely for an individual to be diagnosed as Asperger's and classified in "severely autistic".

We thank the Reviewer for this observation. Behavioral assessment was carried out using CARS which allow an objective and quantifiable assessment of abnormal behaviors in ASD. The CARS was developed to identify and classify children with autism and to determine symptom severity through quantifiable ratings based on direct observation. CARS interrater reliability is good from 0.55 to 0.93, and for validity, the CARS yielded results consistent with the judgments of clinical experts. For better highlighting this characteristic of the scale we modified the text and add  sentence:  “Partecipants”- line 124-125.  

No data regarding individuals' language ability level reported. Although on lines 114 and 115 it is cited that "global Intellectual Quotient (IQ), a non-verbal IQ (performance IQ) and a verbal IQ to be obtained for all ASD children". This might be an indicator that the entire recruited sample was verbal, hence this is wreath to be mentioned clearly that the sample was all verbal. If the entire recruited sample were verbal, there will be a very big question will arise regarding the level 2 and 3 of individuals with ASD that are generally nonverbal or at least reluctant to talk.

We thank the reviewer for this observation. In our preliminary statistical analyses, VQ as nVQ didn't have correlations with others clinical and anthropometric variables, so we highlighted VQ only in Table 1 and the Global IQ' correlations in Table 3. Now we have added correlations about VQ (and n-VQ) also in Table 3.

I was also very curious to know if any individual excluded because of the exclusion criteria form the potential population.  Because on line 148, it is also cited that some individuals were excluded (nor mentioned how many) because they entered into the puberty period which indicates that the sample was bigger than what is reported. Also, online number 160, it is cited that Neurological abnormality was also considered as the exclusion criteria, and based on the Touwen examination some individuals classified as neurologically abnormal but no numbers were presented. It also indicated that the originally recruited sample was much bigger than the final 33. The dropout rate also will be very useful to report. I could not figure out the sample recruitment approach in the methods section, I speculate a convenient sampling approach considered. It needed to be mentioned although if any other approach for sample recruitment is followed.

We thank very much the Reviewer’s perceptive comment which helped us to improve the reported information on recruitment criteria.  We modified the internal structure of “Participants” paragraph adding a flow-chart diagram for better highlighting the inclusion/exclusion criteria.

I also found that individuals with ASD have been addressed as "patient(s)" in this paper several times; this is not in line with available literature in this field. Except for line 89 in which I recommend Sample to be used as a substitution, I suggest substituting individuals /people with ASD instead of ASD patients (line 89, 110, 315, and 132).

We agree with Reviewer’s suggestion and we modified the word “patient(s)” with “partecipants” and  “children”.

I also suggest the “social brain functional morphology” which is mentioned in the abstract and the final line of the discussion part to be explained in paragraph at the Methods section.

We thank the Reviewer for his suggestion. The concept of “social brain” functional (dys)morphology was preliminary highlighted in some previous researches  as Cheung C. et al 2011 or our study Tripi et al 2019.  It is a concept that involves the study of relationships between the structure of an organism (in this case amygdala, part of the social brain structure) and the function of the various parts of an organism (supplementary motor areas). It is a clinical concept and we agree with the reviewer that it would be necessary a better explication, not only a mentioning in the final part of discussion. Then we have added a new sentence in the middle part of Discussion-Line 324.  

I was also curious to know if the anthropometric norms for different ethnicity might differ from what is reported to be as a norm for the Caucasian or at least some explanation about any other available norms for the other ethnical group.

We thank the Reviewer for his observation. The sample consisted of Caucasian children of European origin, as normative physical data have been well-defined only  in the Caucasian population. The sentence is in Line 96-98.

In sum, I understand the challenges of doing a multidisciplinary study such as what is reported here and I know how much work goes into it. Meanwhile, I also worry about the possible misinterpretations of findings of such small scale studies on a convenient sample and a special ethnical group of children with a condition in which very little is known about its causes and etiology.  Therefore, as a safeguard, I do recommend the study to be classified as a preliminary one in title to provide a base for more robust studies in terms of sample size and coverage: “A preliminary study on Cranio-Facial Characteristics associated with Minor Neurological Dysfunctions (MNDs) in Children with Autism Spectrum Disorders (ASD)”.

We agree with Reviewer’s suggestion. The title has been modified.

* English language and style of the manuscript was revised by Mrs Suzanne Heron-Proof reading

Round 2

Reviewer 2 Report

This updated version is an improver version of the manuscript. The authors have done their best to address all the comments and previously raised issues. This is much appreciated.